# MxMoE: Mixed-precision Quantization for MoE with Accuracy and Performance Co-Design

**Haojie Duanmu** [1 2]   **Xiuhong Li** [3]   **Zhihang Yuan** [3]   **Size Zheng** [4]   **Jiangfei Duan** [5]
**Xingcheng Zhang** [2]   **Dahua Lin** [2 5 6]

## Abstract

Mixture-of-Experts (MoE) models face deployment challenges due to their large parameter counts and computational demands. We explore quantization for MoE models and highlight two key insights: 1) linear blocks exhibit varying quantization sensitivity, and 2) divergent expert activation frequencies create heterogeneous computational characteristics. Based on these observations, we introduce MxMoE, a mixed-precision optimization framework for MoE models that considers both algorithmic and system perspectives. MxMoE navigates the design space defined by parameter sensitivity, expert activation dynamics, and hardware resources to derive efficient mixed-precision configurations. Additionally, MxMoE automatically generates optimized mixed-precision Group-GEMM kernels, enabling parallel execution of GEMMs with different precisions. Evaluations show that MxMoE outperforms existing methods, achieving 2.4 lower Wikitext-2 perplexity than GPTQ at 2.25-bit and delivering up to 3.4× speedup over full precision, as well as up to 29.4% speedup over uniform quantization at equivalent accuracy with 5-bit weight-activation quantization. Our code is available at https://github.com/cat538/MxMoE.

## 1. Introduction

Mixture-of-Experts (MoE) architectures have established themselves as a cornerstone of modern large language models, driving state-of-the-art performance across diverse AI tasks (Jiang et al., 2024; The Mosaic Research Team, 2024;

[1]Shanghai Jiao Tong University [2]Shanghai AI Laboratory [3]Peking University [4]ByteDance Seed [5]The Chinese University of Hong Kong [6]CPII under InnoHK. Correspondence to: Haojie Duanmu <duanmuhaojie@sjtu.edu.cn>.

*Proceedings of the 42nd International Conference on Machine Learning*, Vancouver, Canada. PMLR 267, 2025. Copyright 2025 by the author(s).

Yang et al., 2024; Muennighoff et al., 2024; Liu et al., 2024b).These models replace dense MLP blocks in dense Large Language Models (LLMs) with specialized MoE components, each containing a routing mechanism and multiple expert networks. Through dynamic token-to-expert allocation during inference, MoE architectures achieve superior model capacity while maintaining computational efficiency equivalent to their dense counterparts. However, these performance advantages introduce critical deployment challenges that existing hardware systems struggle to address. First, the memory footprint of MoE model is usually several times that of Dense model. For instance, DeepSeek-V3's 671B parameters exceed the memory capacity of eight H100 GPUs in standard configurations (Liu et al., 2024b). Second, while the architecture activates only a subset of experts per token during inference, scenarios such as prefill phase or large-batch serving can trigger widespread expert activation, resulting in considerable computational overhead (Kim et al., 2022a).

Mixed-precision quantization, which allocates different bitwidths to different parts of the model based on certain criteria, has been shown to enhance MoE model performance (Huang et al., 2024; Tang et al., 2024). However, the added complexity of mixed-precision schemes often leads to increased system overhead, making it more challenging to achieve tangible improvements in wall-clock time (Dettmers et al., 2022).

In this paper, we focus on achieving a mixed-precision scheme that balances both model accuracy and hardware efficiency. Our goal is to enhance MoE model quantization while simultaneously achieving meaningful acceleration. To this end, we first conduct an experimental analysis of the MoE block, leading to two key insights: ❶ There is a significant variation in quantization sensitivity across different linear blocks within an MoE block. Unlike previous work (Li et al., 2024; Huang et al., 2024), our findings suggest that allocating bitwidth at the linear block level, rather than the expert level, may yield better results. ❷ The activation frequencies of different experts exhibit large variance within an MoE block. From a hardware perspective, this variance implies that different computational characteristics

(e.g., memory-bound and compute-bound) coexist within an MoE block, indicating potential benefits from applying diverse quantization schemes tailored to these distinct computational characteristics.

Building on these insights, we propose MxMoE, a framework designed to derive an optimal mixed-precision quantization strategy through accuracy and performance co-design. MxMoE first navigates the interplay among parameter sensitivity, expert activation patterns, and hardware resources, optimizing within this multidimensional design space with other constrains to identify the most effective mixed-precision quantization schemes. Furthermore, MxMoE automatically generates a GPU kernel tailored to the identified strategies, efficiently orchestrating linear blocks with different precision in a parallel manner to exploit hardware.

## 2. Background

### 2.1. Quantization

Quantization aims to map a continuous value $x$ to a discrete set of values. This can be represented as: $\hat{x} = \mathcal{Q}(x)$, where $\mathcal{Q}(\cdot)$ represents the quantization function, which can be either non-uniform or uniform. We focus on uniform quantization where the set of discrete values is evenly spaced. The uniform min-max quantization operation can be expressed as:

$$\hat{x} = \text{round}\left(\frac{x - x_{\min}}{\Delta}\right) \cdot \Delta + x_{\min}$$

where $x_{\min}$ is the minimum value of the quantization range, $\Delta$ is the quantization step size, and the rounding function applies rounding to target domain, which is the source of the quantization error.

LLM quantization mainly focuses on weight-only or weight-activation quantization. Weight-only methods (Frantar et al., 2022; Lin et al., 2024a; Kim et al., 2023) only involves quantization of model weight. Weight-activation methods (Dettmers et al., 2022; Xiao et al., 2022) quantize both model weight and intermediate activation. In this work, we focus on MoE block quantization, covering both methods. Another lines of work, KV cache quantization (Liu et al., 2024c; Hooper et al., 2024; Yue et al., 2024; Duanmu et al., 2024) is orthogonal to our work.

### 2.2. MoE Mechanism and Group-GEMM

In MoE models, the routing mechanism dynamically selects a subset of $E$ experts for each input token's hidden state, assigning corresponding weights $\{w_e\}_{e=1}^{E}$. For the $e$-th expert, the computation is given by:

$$W_{\text{down}}^e \left(\sigma(W_{\text{gate}}^e(X_e)) \odot W_{\text{up}}^e(X_e)\right) \quad (1)$$

where $W_{\text{gate}}^e$, $W_{\text{up}}^e$, and $W_{\text{down}}^e$ are linear transformations, $\sigma$ represents the activation function and $\odot$ represents element-wise multiplication. The final output $F$ is the weighted sum of the outputs from all experts:

$$F = \sum_{e=1}^{E} W_{\text{down}}^e \left(\sigma(W_{\text{gate}}^e(X_e)) \odot W_{\text{up}}^e(X_e)\right) \odot w_e \quad (2)$$

A straightforward method for executing such operation on hardware is sequential execution, where the summation in Eq. 2 is expanded, and each expert is processed individually before aggregating the results. However, since the computation independence, this operation can be parallelized. Notably, the shape of the input $X_e$ and the corresponding weight matrix may vary across different experts. The approach that concurrently processes multiple independent General Matrix Multiply (GEMM) with different shapes is referred to as Group-GEMM. Unlike Batched-GEMM, where sub-problems have exactly the same shape, Group-GEMM deals with sub-problems that have varying dimensions, requiring a more careful design. CUTLASS (Thakkar et al., 2023) provides a high-performance implementation capable of efficiently handling the parallel execution of independent GEMM operations.

## 3. Motivation

### 3.1. Heterogeneous Quantization Sensitivity

Recent studies have demonstrated that neural network components exhibit heterogeneous sensitivity to bitwidth, with quantization affecting different parameters to varying extents (Wang et al., 2019; Dong et al., 2019). This heterogeneous sensitivity can be leveraged through mixed-precision, where different bitwidths are allocated to parameters based on their sensitivity. Such schemes typically outperform uniform-precision quantization in terms of accuracy.

In the context of MoE models, several works have investigated the behavioral differences among experts. These studies show that, due to the influence of training dynamics, not all experts are equal. Some experts specialize in specific tokens, contributing less to the overall generation (Liu et al., 2024b; Xue et al., 2024). Building on this idea, we extend the concept of heterogeneity to the quantization of MoE models. Specifically, we systematically investigate quantization sensitivity across different architectural dimensions of MoE models by analyzing the sensitivity of various experts and their corresponding linear blocks.

As illustrated in Fig. 1a, our analysis reveals two key struc-

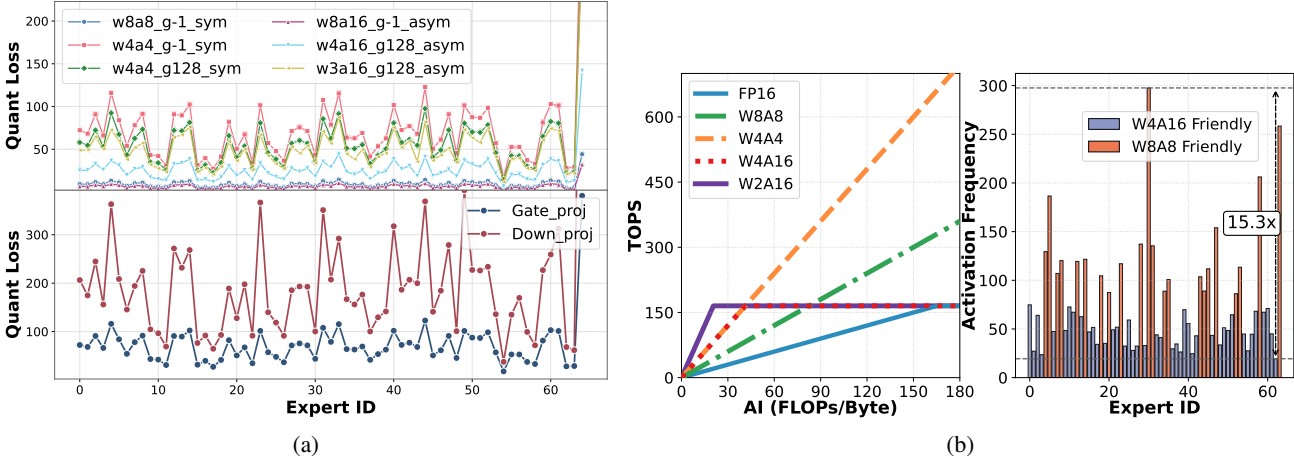

(a)                                                                                      (b)

*Figure 1.* (a) Quantization loss across experts in DeepSeekV2-Lite's 11th layer under various quantization schemes (top), and across linear components (Gate_proj/Down_proj) under the `w4a4_g-1_sym` configuration (bottom). The quantization notation `wxay_gz_b` denotes x-bit weights, y-bit activations, group size z (-1 indicates per-channel/token) with symmetric (sym) or asymmetric (asym) quantization. The quantization loss metric is formally defined in Section 4.2.1. (b) Roofline performance analysis for RTX 4090 GPU (left) and expert activation frequency distribution in DeepSeekV2-Lite's 11th layer (right).

tural patterns. First, experts exhibit divergent sensitivity profiles: for example, Expert 40 suffers significantly greater performance degradation under quantization compared to Expert 37. Second, sensitivity varies considerably across the components within a single expert: the Down_proj block in Expert 40 requires higher precision than the Gate_proj block within the same expert.

These observations motivate our linear-block granularity strategy: assigning different bitwidth to linear-blocks in MoE blocks to preserve model accuracy. Unlike recent studies that adopt expert-level mixed-precision schemes (Li et al., 2024; Huang et al., 2024), our approach focuses on allocating bitwidths at the linear block level. In Section 5.4, we demonstrate the superiority of this linear-block-level allocation strategy. Another line of recent research explores fine-grained mixed-precision approaches at the channel or element level (Kim et al., 2023; Zhao et al., 2024). However, these approaches incur significant computational overhead due to irregular memory access patterns and the need for bitwidth lookup operations (Dettmers et al., 2022).

### 3.2. Hardware Friendly Quantization

The computational efficiency of different quantization schemes varies depending on the specific characteristics of the computation (Lin et al., 2024b). The effectiveness of these schemes is fundamentally determined by the arithmetic intensity (AI), defined as the ratio of FLOPs to memory access in bytes (Williams et al., 2009). Weight-only quantization mitigates memory bandwidth limitations by reducing data transfer, whereas weight-activation quantization leverages low-precision arithmetic units to accelerate compute-intensive operations (Frantar et al., 2024). Our

roofline analysis on the Nvidia RTX-4090 (Fig. 1b) identifies distinct performance regimes: for GEMM operations with shape $[m, n, k]$ where $n, k \gg m$, the arithmetic intensity simplifies to $\mathcal{A} = m$. For example, our analysis shows that W4A16 outperforms W8A8 when $\mathcal{A} < 83$ and W2A16 outperformes W4A4 when $\mathcal{A} < 42$.

In addition, we observe that MoE architectures exhibit significant computational heterogeneity. For instance, our evaluation of DeepSeekV2-Lite on the HumanEval-X dataset (Zheng et al., 2023) reveals that expert activation frequencies within individual MoE blocks vary by over $10\times$ (Fig. 1b). Considering W8A8 and W4A16, the computational heterogeneity implies that within a single MoE block, operations that are suited for W8A8 and W4A16 coexist simultaneously, as predicted by the roofline model. This characteristic, distinct from dense LLMs, suggests that by strategically combining quantization schemes across experts, we can potentially achieve better performance than uniform precision quantization.

From a hardware-friendly perspective, it is possible to select the most efficient quantization schemes based on computational characteristics. For example, W2A16 generally outperforms W4A16, and W4A4 outperforms W8A8. However, as discussed in Section 3.1, the allocation of bitwidth plays a critical role in model accuracy. Simply optimizing for performance may degrade model accuracy, while focusing exclusively on accuracy can result in suboptimal performance.

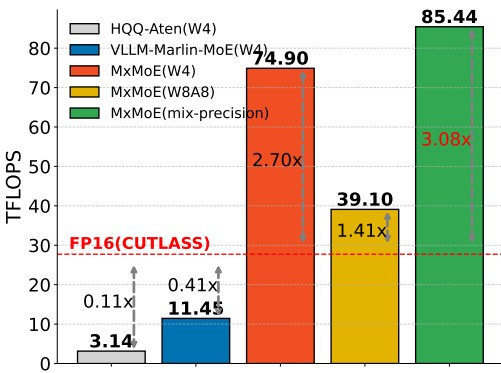

*Figure 2.* Comparison of the computation throughput of low-precision MoE block. W4 denotes 4-bit per-channel symmetric weight-only quantization, while W8A8 refers to 8-bit per-channel symmetric weight-activation quantization. The problem consists of 60 experts, each with a shape of $[N, K] = [2816, 2048]$ (from Qwen2_MoE1.5), with each token activating 4 experts. The total number of input tokens is set to 512.

### 3.3. Algorithm-System Co-Design and Challenges

The analysis presented raises a fundamental question: can we design a quantization scheme specifically tailored to MoE models that effectively balancing model accuracy and computational efficiency? Our findings suggest that heterogeneous quantization sensitivity at the linear-block level within MoE models significantly affects accuracy, while hardware resources determine the maximum achievable computational efficiency. Moreover, the variation in expert activation frequencies introduces divergent computational demands, which is crucial for identifying the optimal quantization strategy. Therefore, an effective mixed-precision quantization scheme must take into account three key factors: 1) parameter sensitivity, 2) expert activation frequencies, and 3) hardware characteristics.

For a given mixed-precision scheme, system-level support is necessary to translate theoretical performance improvements into actual wall-clock time reductions. While numerous works have optimized low-precision operators for dense LLMs (Zhao et al., 2024; Lin et al., 2024b), these approaches are often ill-suited for the MoE models. To demonstrate this, we leverage two widely used low-precision kernels: HQQ and VLLM-Marlin-MoE to build a MoE blocks with 4-bit weight, as shown in Fig. 2. HQQ, which does not fuse dequantization, significantly underperforms the full-precision baseline. Marlin (Frantar et al., 2024) is a highly optimized W4A16 kernel achieves SOTA performance for W4A16 GEMM. The VLLM community (Kwon et al., 2023) adopts Marlin to build the VLLM-Marlin-MoE kernel. It sequentially invokes the Marlin kernel multiple times for each expert, which results in suboptimal GPU utilization. These shortcomings intensify when introducing mixed-precision configurations, as existing kernel designs lack the architec-

tural flexibility to handle precision-heterogeneous expert computations efficiently.

We propose MxMoE to address above challenges. MxMoE tightly couple **1)** a hardware-aware bitwidth allocation scheme that respects parameter sensitivity and activation patterns with **2)** a specialized computation engine that eliminates kernel launch overhead and enables parallel mixed-precision expert execution.

## 4. Method

### 4.1. Overview

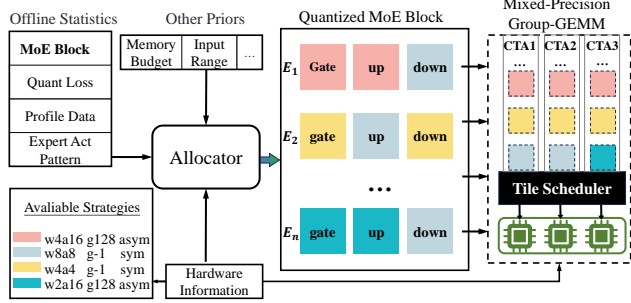

*Figure 3.* Overview of MxMoE.

The workflow of MxMoE is illustrated in Fig. 3. ❶ First, MxMoE's allocator takes statistical data specific to the MoE model as input, navigating the interplay between parameter sensitivity, expert activation patterns, and hardware resources. It then optimizes within this multidimensional design space to identify the mixed-precision quantization scheme. ❷ Next, MxMoE generates a mixed-precision Group-GEMM kernel tailored to the identified scheme, efficiently orchestrating linear blocks with varying precision. During runtime, the automatically generated tile scheduler maps mixed-precision computation tasks to hardware in a load-balanced manner, fully parallelizing the MoE block.

We begin by formalizing the impact of quantization and expert activation frequency on model accuracy and execution performance, providing a comprehensive understanding of the design space for mixed-precision scheme. Subsequently, we present our solution and discuss the system-level support required for mixed-precision MoE blocks.

### 4.2. Hardware-Aware Bitwidth Allocation

For a Given $M$-layer MoE model with parameter $W$ and input $X$, the objective of bitwidth allocation in MxMoE is to minimize both the perturbation introduced by quantization and the total execution time of all MoE blocks in the model:

$$\min(\mathcal{L}(W, X) - \mathcal{L}(W_q, X_q))^r \cdot (\sum_{i}^{M} T_i)^{1-r} \quad (3)$$

where $r$ is a hyper-parameter balancing the trade-off between model accuracy loss and execution time. In this study, we adopt the setting presented in (Choukroun et al., 2019) which assumes a positive correlation between the change in the intermediate output of the quantized model and the final output. Therefore, minimizing the intermediate output loss leads to minimize the loss item in Eq.3. Furthermore, since the model is executed sequentially, minimizing the execution time of individual MoE blocks contributes directly to reducing the overall execution time item in Eq.3. Thus, the objective simplifies to minimizing the output loss $L$ and execution time $T$ of a single mixed-precision MoE block:

$$\min L^r \cdot T^{1-r} \tag{4}$$

To further detail the formulation, we decompose the terms $L$ and $T$ systematically.

### 4.2.1. QUANTIZATION LOSS FORMULATION

Let $S$ denote the set of hardware-supported quantization schemes (e.g., W6A6 is still unsupported by most existing hardware, while FP8 is supported on Nvidia RTX-4090 but not A100). For an MoE block comprising $E$ experts, each containing $N$ linear blocks (typically $N = 3$ for modern architectures, corresponding to gate_proj, up_proj, and down_proj). The composite loss aggregates individual quantization effects as:

$$L = \sum_{i=1}^{E} \sum_{j=1}^{N} \sum_{k=1}^{|S|} \Delta_{i,j,k} \cdot x_{i,j,k} \tag{5}$$

where $x_{i,j,k} \in \{0,1\}$ denotes the binary selection variable for applying the $k$-th quantization scheme to the $j$-th linear block in expert $i$. The perturbation coefficient $\Delta_{i,j,k}$ quantifies the output distortion when using scheme $k$, computed via Euclidean distance between full-precision ($\mathbf{O}$) and partially quantized ($\hat{\mathbf{O}}$) MoE block outputs:

$$\Delta = \left\| \hat{\mathbf{O}} - \mathbf{O} \right\|_2 \tag{6}$$

For practical estimation, we employ a small calibration set (e.g., 128 samples from WikiText2) to compute $\Delta_{i,j,k}$ values. Each linear block in expert $i$ is sequentially quantized with scheme $k \in S$, with the corresponding output perturbation statistically estimated across calibration samples.

### 4.2.2. RUNTIME COST MODELING

To model the execution time $T$ of mixed-precision MoE blocks, we first analyze individual linear blocks. Given input token distributions and expert activation frequencies, we derive GEMM shapes for each linear-block based on per-expert token allocations. On modern GPUs, GEMM

is decomposed into multiple sub-problems, known as tiles, mapped to SMs for parallel execution. MxMoE generates candidate tile configurations for each quantization scheme and profiles their runtime costs $c_t$ ahead-of-time. For a linear block with scheme $s$, tile decomposition is represented as $(c_t, n_t)$, where $n_t$ denotes the tile count.

The computational task of a MoE block can be represented as a list of such pairs, and our goal is to estimate the execution time for this set of tasks mapped to the GPU. In our system design, all tiles are parallelized across the SMs, and the total execution time of the MoE block depends on the longest execution time across all SMs: $T = \max_{i=1}^{P} T_i$ where $P$ is the number of SMs. However, the use of the maximum operator complicates the optimization procedure. To address this, we approximate based on the observed fact: the number of tiles decomposed from a MoE block substantially exceeds the number of SMs. Therefore, the execution time of the entire MoE block can be approximated as the serial execution time of all tiles, divided by the number of SMs $P$:

$$T = \frac{1}{P} \sum_{i=1}^{E} \sum_{j=1}^{N} \sum_{k=1}^{|S|} \sum_{t=1}^{|\mathcal{T}|} c_{i,j,k,t} \cdot y_{i,j,k,t} \cdot x_{i,j,k}$$

where $\mathcal{T}$ represents candidate tile configurations, $c_{i,j,k,t}$ denotes execution time of linear-block $(i, j)$ under scheme $k$ with tile configuration $t$, and $y_{i,j,k,t} \in \{0,1\}$ is indicating variable for selecting tile configurations. Our experiments show that this approximation is effective because the total number of tiles is typically much larger than the number of SMs.

The optimization under certain memory budget $M$ is formulated as an ILP problem:

$$\text{MINIMIZE} \quad L^r \cdot T^{(1-r)}$$

$$\text{WHERE} \quad L = \sum_{i=1}^{E} \sum_{j=1}^{N} \sum_{k=1}^{|S|} \Delta_{i,j,k} \cdot x_{i,j,k}$$

$$\text{WHERE} \quad T = \frac{1}{P} \sum_{i=1}^{E} \sum_{j=1}^{N} \sum_{k=1}^{|S|} \sum_{t=1}^{|\mathcal{T}|} c_{i,j,k,t} \cdot y_{i,j,k,t} \cdot x_{i,j,k}$$

$$\text{SUBJECT TO} \quad x_{i,j,k} \in 0,1, \quad \sum_{k=1}^{|S|} x_{i,j,k} = 1, \quad \forall i,j$$

$$\text{SUBJECT TO} \quad y_{i,j,k,t} \in 0,1, \quad \sum_{t=1}^{|\mathcal{T}|} y_{i,j,k,t} = 1, \quad \forall i,j,k$$

$$\text{SUBJECT TO} \quad \sum_{i=1}^{E} \sum_{j=1}^{N} \sum_{k=1}^{|S|} W_{i,j,k} \cdot x_{i,j,k} \leq M \tag{7}$$

We first gather offline statistics $\Delta$ and expert activation patterns. The execution time $c$ for each linear block, under every quantization scheme and tile configuration, is then estimated based on pre-profiled single-tile runtime costs. This ILP is subsequently solved to determine desired mixed-precision schemes and tile configurations while ensure the quantized weight strictly adhere the memory budget. Following quantization scheme allocation, we apply randomized Hadamard transformations to model weights using the incoherence processing used in QuaRot (Ashkboos et al., 2024), then perform GPTQ-based quantization (Frantar et al., 2022). Activations are dynamically quantized at runtime according to the corresponding allocated scheme.

### 4.3. Mixed-Precision GEMM Orchestration

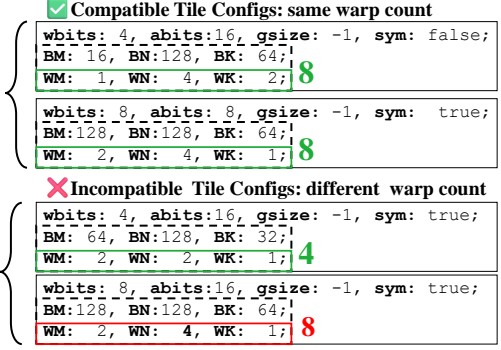

*Figure 4.* MxMoE ensures that tile configurations for different precisions have the same number of warps.

As discussed in Section 3.3, sequentially processing computations for each expert is inefficient. CUTLASS does not support heterogeneous precisions. Fusing heterogeneous-precision GEMMs introduces two fundamental challenges: **1)** Optimal tile sizes and warp layouts vary across precisions, making a unified kernel for all possible precisions inherently suboptimal, and **2)** The large number of possible precision combinations makes developing a custom kernel for each combination prohibitively costly. MxMoE addresses these challenges through an automated kernel generation framework that consists of three key components: micro-kernel specialization, resource configuration, and tile scheduling.

**Micro-Kernel Specialization.** We introduce configurable CTA-level micro-kernels implemented as CUDA device functions, designed with Cooperative-Thread-Group (CTA) index independence to enable subsequent horizontal fusion. Each micro-kernel's resources are specified via C++ template parameters, while memory access patterns are optimized for specific quantization schemes through meticulous hand-tuning of compute-to-memory access pipelines.

For instance, the W2A16 micro-kernel integrates fused de-

quantization with bit manipulation techniques for optimized integer-to-float conversion (Kim et al., 2022b), while the W4A4-g128 variant employs multistage software pipelining that enforces strict adherence to 128 quantization group constraints. Following bitwidth allocation, MxMoE generates a tile scheduler with a precision-aware routing logic, composing heterogeneous GEMM operations into unified kernel execution streams. We discuss in detail the advantages of micro kernel specilization over other possible approaches in App. A.2.

**Resource Configuration.** We next address the configuration of computational resources for horizontally fused mixed-precision Group-GEMM kernels. Building on the hardware-aware bitwidth allocation in Eq. 7, MxMoE derives tile configurations optimized for each quantization method under two critical constraints. First, warp count consistency is enforced across all micro-kernel tile configurations as shown in Fig. 4. Second, shared memory allocation follows the maximum requirement among fused operations. These two constraints ensures compliance with the CUDA programming model's requirement for uniform resources across CTAs (as shown in Fig. 3).

To mitigate shared memory waste from divergent tile sizes, we employ k-dimension tiling (slice-K). As illustrated in Fig. 4, the tile size for W4A16 is substantially smaller than that of W8A8. This disparity results in shared memory under-utilization for W4A16 micro-kernel. Our solution introduces additional parallelism along the k-dimension for W4A16 configurations through strategic tile partitioning. This dual-purpose optimization simultaneously reduces warp under-utilization while increasing shared memory utilization.

**Tile Schedule.** Finally, MxMoE optimizes the scheduling of tiles with heterogeneous precision requirements. The execution time of tiles varies significantly across different precision and tile shape configuration, making the scheduling order a critical determinant of overall completion time. This is a classic makespan minimization problem. While dynamic programming can achieve optimal solution, MxMoE implements an efficient greedy heuristic that prioritizes computationally intensive tiles. Given that the number of tiles in MoE blocks typically exceeds the available SM count by a substantial margin, this approach achieves near-optimal performance (Graham, 1966) while significantly reducing the scheduling overhead compared to dynamic programming solutions.

## 5. Experiments

### 5.1. Experimental Setup

**Model Configurations.** We evaluate MxMoE on three open-source MoE architectures: DeepSeek (Liu et al., 2024a),

*Table 1.* We evaluate on the following datasets: Arc-Challenge (AC), Arc-Easy (AE), HellaSwag (HS), LAMBADA-openai (LO), LAMBADA-standard (LS), PIQA (PQ), and WinoGrande (WG). GPTQ⋆ denotes GPTQ with random Hadamard transformation preprocessing. #Bits indicates the quantization bitwidth for GPTQ and Quarot (uniform bitwidth), and the average bitwidth for MxMoE.

| Model | Method | #Bits (W-ACT) | AC↑ | AE↑ | HS↑ | LO↑ | LS↑ | PQ↑ | WG↑ | Avg.↑ | PPL↓ |
|---|---|---|---|---|---|---|---|---|---|---|---|
| | Baseline | 16-16 | 48.98 | 76.22 | 77.91 | 72.33 | 67.90 | 80.20 | 71.19 | 70.68 | 5.92 |
| DeepSeekV2-Lite | GPTQ⋆ | 3.25-16 | 47.35 | 75.04 | 76.44 | 70.41 | 65.65 | 79.05 | 71.27 | 69.32 | 6.18 |
| | GPTQ⋆ | 2.25-16 | 37.63 | 63.47 | 65.45 | 52.53 | 48.55 | 74.59 | 64.09 | 58.04 | 8.49 |
| | QuaRot | 4-4 | 41.81 | 67.51 | 74.12 | 50.01 | 45.86 | 75.52 | 63.38 | 59.74 | 8.44 |
| | MxMoE | 3.25-16 | 47.87 | 74.58 | 76.85 | 71.10 | 65.85 | 79.27 | 70.09 | 69.37 | 6.08 |
| | MxMoE | 2.25-16 | 40.36 | 68.86 | 68.63 | 59.56 | 54.01 | 75.08 | 67.80 | 62.04 | 7.01 |
| | MxMoE | 5-5 | 46.76 | 74.37 | 77.38 | 68.41 | 64.99 | 79.38 | 69.22 | 68.64 | 6.16 |
| | Baseline | 16-16 | 44.03 | 69.53 | 77.26 | 71.28 | 64.62 | 80.47 | 69.30 | 68.07 | 6.79 |
| Qwen1.5-MoE | GPTQ⋆ | 3.25-16 | 43.34 | 68.60 | 75.35 | 68.68 | 62.80 | 79.22 | 66.54 | 66.36 | 7.15 |
| | GPTQ⋆ | 2.25-16 | 30.89 | 47.14 | 60.77 | 43.72 | 34.81 | 69.97 | 56.20 | 49.07 | 11.19 |
| | QuaRot | 4-4 | 27.13 | 40.74 | 57.10 | 35.61 | 25.33 | 66.43 | 51.93 | 43.47 | 18.44 |
| | MxMoE | 3.25-16 | 43.77 | 66.04 | 75.92 | 69.71 | 62.82 | 79.11 | 68.03 | 66.49 | 7.02 |
| | MxMoE | 2.25-16 | 31.66 | 53.28 | 62.80 | 56.43 | 51.00 | 71.33 | 61.25 | 55.39 | 8.79 |
| | MxMoE | 5-5 | 42.92 | 66.04 | 76.27 | 70.06 | 63.40 | 80.58 | 67.80 | 66.72 | 7.01 |
| | Baseline | 16-16 | 55.20 | 77.19 | 84.09 | 74.35 | 62.62 | 82.32 | 72.14 | 72.56 | 5.84 |
| Qwen2-MoE | GPTQ⋆ | 3.25-16 | 53.67 | 75.88 | 82.90 | 73.36 | 63.24 | 81.01 | 70.96 | 71.57 | 6.11 |
| | GPTQ⋆ | 2.25-16 | 38.82 | 57.66 | 71.27 | 58.99 | 49.72 | 73.29 | 60.30 | 58.58 | 7.98 |
| | QuaRot | 4-4 | 33.19 | 42.72 | 54.34 | 23.02 | 9.53 | 63.87 | 50.12 | 39.54 | 110.66 |
| | MxMoE | 3.25-16 | 53.84 | 76.30 | 82.81 | 72.39 | 60.95 | 81.34 | 69.69 | 71.05 | 6.18 |
| | MxMoE | 2.25-16 | 45.05 | 68.86 | 77.13 | 66.00 | 56.61 | 75.41 | 62.90 | 64.57 | 7.57 |
| | MxMoE | 5-5 | 54.86 | 75.55 | 82.69 | 72.87 | 62.68 | 79.49 | 70.96 | 71.30 | 6.25 |
| | Baseline | 16-16 | 66.38 | 85.39 | 85.95 | 77.28 | 73.06 | 85.20 | 76.72 | 78.57 | 3.88 |
| Mixtral-8×7B | GPTQ⋆ | 3.25-16 | 64.42 | 84.01 | 85.12 | 76.77 | 71.76 | 83.79 | 76.16 | 77.43 | 4.17 |
| | GPTQ⋆ | 2.25-16 | 48.89 | 72.35 | 76.95 | 68.39 | 61.44 | 77.15 | 67.72 | 67.56 | 5.69 |
| | QuaRot | 4-4 | 50.60 | 68.69 | 75.65 | 40.95 | 38.83 | 76.88 | 61.01 | 58.94 | 9.06 |
| | MxMoE | 3.25-16 | 64.25 | 84.22 | 85.04 | 76.98 | 71.86 | 84.17 | 75.93 | 77.49 | 4.15 |
| | MxMoE | 2.25-16 | 48.98 | 72.77 | 77.44 | 68.68 | 62.18 | 76.28 | 68.90 | 67.89 | 5.63 |
| | MxMoE | 5-5 | 64.08 | 83.71 | 85.10 | 76.21 | 71.78 | 83.79 | 73.80 | 76.92 | 4.20 |

Mixtral (Jiang et al., 2024), and Qwen (Yang et al., 2024) as detailed in Table 2. DeepSeek-V2-Lite employs a hybrid architecture: the first layer employs dense MLP instead of MoE blocks in other layers which are quantized with GPTQ 4-bit per-channel asymmetric quantization in our experiments. We focus on the quantization of MoE blocks, retaining full precision for the attention modules. Experiments conducted on Nvidia RTX-4090.

*Table 2.* Architectural Specifications of Evaluated MoE Models

| Model Variant | Params (GB) | Experts | TopK |
|---|---|---|---|
| Mixtral-8×7B-Instruct-v0.1 | 92.9 | 8 | 2 |
| Qwen1.5-MoE | 26.7 | 60+4 | 4 |
| Qwen2-MoE-Instruct | 106.9 | 64+8 | 8 |
| DeepSeek-V2-Lite | 29.3 | 64+2 | 6 |

**Calibration.** MxMoE requires offline calibration to determine the sensitivity of linear-block and expert activation frequencies for bitwidth allocation. For all experiments, we use 128 sequences, each of length 4096, drawn from the Wikitext2 training set (Merity et al., 2016). This calibration process typically takes from several minutes to a few hours depending on the model size. To enhance quantiza-

tion accuracy, we apply a random hadamard transformation. We disabled online rotations because they failed on some models like DeepSeek-V2-Lite due to the shape constrain. For weight quantization, MxMoE employs GPTQ, using the same calibration set with that of bitwidth allocation.

## 5.2. Accuracy Results

For weight-only quantization we test 3-bit and 2-bit quantization, comparing with GPTQ, configured with group size 128, asymmetric min-max quantization, where the scale and zero-point are stored in 16-bit format, resulting in an average bitwidth of 3.25 and 2.25, respectively. To ensure fairness, we apply the same random Hadamard transformation for GPTQ. MxMoE use $r = 1$ as extremely low-bitwidth implies resource-constrained environment, where model accuracy is more important. The results in Tab. 1 show that GPTQ suffers significant performance degradation at 2.25-bit, while MxMoE consistently outperforms GPTQ across all models. At 3.25-bit, MxMoE outperforms GPTQ in most models except Qwen2-MoE. This exception may stem from inaccuracies in the sensitivity statistics, amplified by inter-layer dependencies (Yue et al., 2024). Using a cross-layer loss as sensitivity metric instead layer loss in Eq. 6

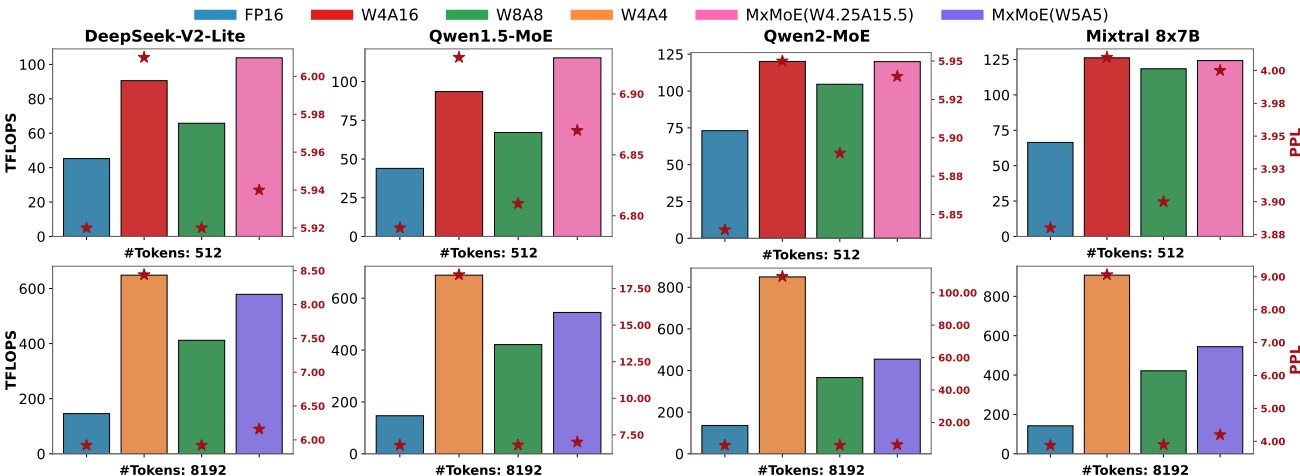

*Figure 5.* Computational throughput of MoE blocks across models and precision settings. W4A16 denotes 4-bit weight-only per-channel asymmetric quantization; W8A8 and W4A4 denote 8/4-bit weight-activation per-channel symmetric quantization. Number followed MxMoE represents the average bitwidth for weight and activation. The ⋆ symbols indicate corresponding perplexity values on WikiText2.

may mitigate this issue, and we leave this to future work. Improvement of MxMoE on Mixtral is relative margin. This is due to Mixtral's fewer experts (Tab. 2), resulting in a more limited mixed-precision design space.

For weight-activation quantization, we compare MxMoE with Quarot (Ashkboos et al., 2024). MxMoE use $r = 0.75$. We focus on 4-bit as 8-bit is almost lossless. Quarot experiences a significant accuracy drop at 4-bit across all models, while MxMoE achieves substantial improvements with just 1 additional bit (i.e. average 5-bit). These gains are attributed to the higher sensitivity of certain activations to bitwidth, as shown in prior studies (Dettmers et al., 2022; Sun et al., 2024). By allocating higher bitwidth for the sensitive activation, MxMoE delivers notable performance improvements. The sources of accuracy gains under this configuration are rigorously dissected in App. A.1.

### 5.3. Performance Analysis

Due to the lack of established low-precision Group-GEMM baselines, we evaluate both uniform-bitwidth and mixed-precision kernels generated by MxMoE. MxMoE use $r = 0.75$ for all the mixed-precision scheme. 16-bit Group-GEMM kernel is from CUTLASS. we assess the computational throughput of MoE blocks across different models, only expert computation is counted as other operations ( gating, topk, sort etc.) is negligible. We randomly sample sequences from WikiText-2 with lengths of 512 and 8192 tokens, respectively representing memory-bound and compute-bound workloads. As shown in Fig. 5, mixed-precision scheme achieve substantial performance improvements: $1.6 - 2.7\times$ throughput increase for memory-bound workloads and $3 - 3.4\times$ for compute-bound workloads compared to full-preicision.

*Table 3.* Comparison of different allocation granularities. Test with 5-bits weight-activation quantization. Evaluation metrics are the same as described in settings.

| Model | PPL↓ | | Avg-Acc↑ | |
|---|---|---|---|---|
| | Linear | Expert | Linear | Expert |
| DeepSeek-V2-Lite | **6.11** | 6.32 | **69.01** | 67.88 |
| Qwen1.5-MoE | **6.95** | 6.98 | **67.35** | 67.11 |

For the memory-bound scenario (512 tokens), W8A8 consistently underperforms both W4A16 and W4.25A15.5, as the limited tokens per expert make MoE block computations memory-bound. MxMoE (W4.25A15.5) not only achieves better accuracy than W4A16 but also delivers up to 25% higher throughput on Qwen1.5-MoE. This improvement stems from our hardware-aware bitwidth allocation strategy, which assigns lower-precision activations to frequently activated experts that form compute-bound operations (as discussed in Section 3.2). In the compute-bound scenario (8192 tokens), we compare against W4A4 and W8A8. While W4A4 offers significant speedup at the cost of substantially increased perplexity, and W8A8 maintains accuracy with minimal acceleration, MxMoE (W5A5) achieves up to 29.4% performance improvement over W8A8 while maintaining comparable perplexity to full-precision models.

### 5.4. Ablation Studies

**Effect of bitwidth allocation granularity.** MxMoE employ linear-block level allocation instead of expert-level allocation in previous studies. We also perform bitwidth allocation at expert level as shown in table 3. The results demonstrate that linear-block allocation consistently outperforms expert-level allocation.

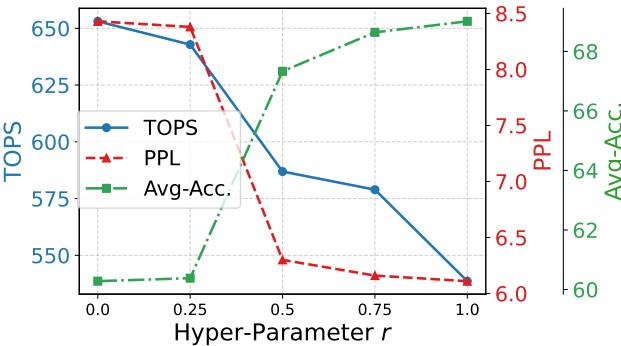

*Figure 6.* Impact of the hyperparameter $r$ on the trade-off between model accuracy and performance. Model: DeepSeek-V2-Lite.

**Impact of the hyperparameter.** MxMoE introduces the hyperparameter $r$ to balance efficiency and accuracy. We employ $r = 0.75$ in all experiments except extremely low-bitwidth weight-only quantization in Tab. 1, where $r = 1$. Intuitively, $r = 1$ prioritizes maximizing accuracy, while $r = 0$ focuses solely on efficiency. Now we quantitatively investigate the impact of the tradeoff parameter. As shown in Fig. 6, performance improves as $r$ decreases, at the cost of reduced accuracy. Notably, when optimizing for both objectives, such as at $r = 0.75$, we observe significant performance gains with minimal accuracy drop. This highlights the effectiveness of hardware-aware bitwidth allocation.

## 6. Conclusion

We propose MxMoE, an accuracy-performance co-design framework for MoE mixed-precision quantization. MxMoE allocates bitwidth through joint optimization of computational efficiency and model accuracy and generates optimized mixed-precision Group-GEMM kernels, achieving significant acceleration while preserving model accuracy.

## Acknowledgements

The authors would like to thank the diligent anonymous reviewers for their constructive feedback. Project supported by Shanghai Municipal Science and Technology Major Project. This study was supported in part by the InnoHK initiative of the Innovation and Technology Commission of the Hong Kong Special Administrative Region Government.

## Impact Statement

This paper presents work whose goal is to advance the field of Machine Learning. There are many potential societal consequences of our work, none which we feel must be specifically highlighted here.

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

# A. Appendix

## A.1. Case Study: Detailed Analysis of MxMoE W5A5 Scheme

We conduct in-depth analysis of MxMoE's W5A5 mechanism and its accuracy advantages. As shown in Tab. 4, reducing activation bitwidth from 5 to 4 bits causes significant model quality degradation, demonstrating critical quantization sensitivity around 4-bit activation precision. This can be attribute to the massive outlier observed in the input activation of Down_proj (Sun et al., 2024), where heavy-tailed activation distributions require higher precision preservation. MxMoE dynamically identifies these quantization-sensitive components (whose bitwidth reduction causes substantial model quality degradation) and allocates elevated bitwidth accordingly. The heterogeneous bitwidth allocation strategy for Qwen1.5-MoE is visualized in Tab. 7.

To further validate mixed-precision benefits, we compare against QuaRot (Ashkboos et al., 2024) in Tab. 5. While uniform bitwidth scaling shows similar perplexity improvement trends in QuaRot, practical deployment remains constrained by the capacity of model hardware, on which 5-bits operation is not supported. In contrast, MxMoE achieves better accuracy while maintaining hardware compatibility through mixed-precision allocation that leverages existing low-bitwidth arithmetic units.

Table 4. WikiText2 perplexity under different weight-activation bitwidth (RTN-token/channel quantization). Column: activation bitwidth, row: weight bitwidth.

| #Bits | 4 | 5 | 6 | 7 | 8 |
|---|---|---|---|---|---|
| 4 | 68079.039 | 41.433 | 11.298 | 9.406 | 8.068 |
| 5 | 12305.585 | 38.707 | 9.715 | 8.169 | 7.335 |
| 6 | 14251.822 | 26.297 | 9.216 | 8.196 | 7.204 |
| 7 | 18151.474 | 34.775 | 9.747 | 8.182 | 7.325 |
| 8 | 19091.917 | 38.990 | 9.525 | 8.260 | 7.278 |

Table 5. WikiText2 perplexity of Qwen1.5-MoE under different quantization bitwidth settings. Both MxMoE and QuaRot employ RTN weight quantization.

| Setting | w4a4 | w5a5 | w6a6 | w7a7 | w8a8 |
|---|---|---|---|---|---|
| QuaRot (Uni) | 36.385 | 7.998 | 6.990 | 6.852 | 6.814 |
| MxMoE (Mix) | - | **7.160** | - | - | - |

## A.2. The Necessity of Automated Kernel-Generation

MxMoE mitigates the combinatorial explosion of mixed-precision configurations by automating kernel generation 4.3. To illustrate the effectiveness of this approach, we compare our strategy with two alternative solutions:

- **Developing a universal kernel to handle all precision combinations**: This approach would compromise kernel performance. We provide a breakdown demonstrating the limitations of this method relative to micro-kernel specialization in MxMoE. Specifically, the kernel for W4A4-per-channel could theoretically share the same software pipeline with W4A4-group128, but enforcing universality significantly degrades performance. As shown in Tab. 6, we test different kernels under the shape [8192, 8192, 8192], and the specialized kernel always outperform unified kernels. The reason is that unifying the two pipelines requires introducing runtime condition checks, which hinder loop unrolling in the MAC-loop. Moreover, to support group-size=128, the per-channel kernel's tile-size selection is constrained, making configurations such as $\text{tile\_k} = 256$ infeasible.

Table 6. Performance comparison of different W4A4 quantization kernels on GPU TOPS

| Kernel Type | W4A4_per-channel TOPS | W4A4_group128 TOPS |
|---|---|---|
| W4A4_per-channel (Specialized) | 1070.5303 | N/A |
| W4A4_group128 (Specialized) | N/A | 667.3349 |
| Unified Kernel | 929.1997 | 412.0268 |

- **Developing separate kernels for each configuration**: While handcrafted kernels could match performance, they require substantial engineering effort. If a given hardware platform supports five quantization candidates (e.g., w2a6, w4a16, w8a8, w4a4, w4a4 with group-size 128), implementing individual kernels for all possible configurations would require $5! = 120$ kernels. In contrast, our micro-kernel specialization approach requires implementing only 5 configurable micro-kernels, which are automatically combined by the kernel generator to form optimized fused operators.

Table 7: W5A5 mixed-precision scheme allocated by MxMoE. Qwen1.5-MoE, layer 5.

| Expert | Gate | | | Up | | | Down | | |
|---|---|---|---|---|---|---|---|---|---|
| | w-act | w_gsize | a_gsize | w-act | w_gsize | a_gsize | w-act | w_gsize | a_gsize |
| 0 | 4-4 | 128 | 128 | 4-4 | 128 | 128 | 4-4 | 128 | 128 |
| 1 | 4-4 | 128 | 128 | 4-4 | 128 | 128 | 8-8 | -1 | -1 |
| 2 | 4-4 | 128 | 128 | 4-4 | 128 | 128 | 8-8 | -1 | -1 |
| 3 | 4-4 | 128 | 128 | 4-4 | 128 | 128 | 8-8 | -1 | -1 |
| 4 | 4-4 | -1 | -1 | 4-4 | -1 | -1 | 4-4 | 128 | 128 |
| 5 | 4-4 | 128 | 128 | 4-4 | 128 | 128 | 4-4 | 128 | 128 |
| 6 | 4-4 | 128 | 128 | 4-4 | 128 | 128 | 8-8 | -1 | -1 |
| 7 | 4-4 | -1 | -1 | 4-4 | -1 | -1 | 4-4 | 128 | 128 |
| 8 | 4-4 | 128 | 128 | 4-4 | 128 | 128 | 8-8 | -1 | -1 |
| 9 | 4-4 | 128 | 128 | 4-4 | 128 | 128 | 8-8 | -1 | -1 |
| 10 | 4-4 | 128 | 128 | 4-4 | 128 | 128 | 8-8 | -1 | -1 |
| 11 | 4-4 | 128 | 128 | 4-4 | 128 | 128 | 8-8 | -1 | -1 |
| 12 | 4-4 | 128 | 128 | 4-4 | 128 | 128 | 8-8 | -1 | -1 |
| 13 | 4-4 | 128 | 128 | 4-4 | 128 | 128 | 4-4 | 128 | 128 |
| 14 | 4-4 | -1 | -1 | 4-4 | -1 | -1 | 4-4 | 128 | 128 |
| 15 | 4-4 | 128 | 128 | 4-4 | 128 | 128 | 8-8 | -1 | -1 |
| 16 | 4-4 | 128 | 128 | 4-4 | 128 | 128 | 8-8 | -1 | -1 |
| 17 | 4-4 | 128 | 128 | 4-4 | 128 | 128 | 8-8 | -1 | -1 |
| 18 | 4-4 | 128 | 128 | 4-4 | 128 | 128 | 8-8 | -1 | -1 |
| 19 | 4-4 | 128 | 128 | 4-4 | 128 | 128 | 8-8 | -1 | -1 |
| 20 | 4-4 | 128 | 128 | 4-4 | 128 | 128 | 4-4 | 128 | 128 |
| 21 | 4-4 | 128 | 128 | 4-4 | 128 | 128 | 8-8 | -1 | -1 |
| 22 | 8-8 | -1 | -1 | 8-8 | -1 | -1 | 8-8 | -1 | -1 |
| 23 | 4-4 | 128 | 128 | 4-4 | 128 | 128 | 4-4 | 128 | 128 |
| 24 | 4-4 | 128 | 128 | 4-4 | 128 | 128 | 8-8 | -1 | -1 |
| 25 | 4-4 | 128 | 128 | 4-4 | 128 | 128 | 4-4 | 128 | 128 |
| 26 | 4-4 | 128 | 128 | 4-4 | 128 | 128 | 8-8 | -1 | -1 |
| 27 | 4-4 | 128 | 128 | 4-4 | 128 | 128 | 4-4 | 128 | 128 |
| 28 | 4-4 | 128 | 128 | 4-4 | 128 | 128 | 8-8 | -1 | -1 |
| 29 | 4-4 | 128 | 128 | 4-4 | 128 | 128 | 4-4 | 128 | 128 |
| 30 | 4-4 | 128 | 128 | 4-4 | 128 | 128 | 4-4 | 128 | 128 |
| 31 | 4-4 | 128 | 128 | 4-4 | 128 | 128 | 8-8 | -1 | -1 |
| 32 | 4-4 | 128 | 128 | 4-4 | 128 | 128 | 4-4 | 128 | 128 |
| 33 | 4-4 | 128 | 128 | 4-4 | 128 | 128 | 4-4 | 128 | 128 |
| 34 | 4-4 | 128 | 128 | 4-4 | 128 | 128 | 8-8 | -1 | -1 |
| 35 | 4-4 | 128 | 128 | 4-4 | 128 | 128 | 8-8 | -1 | -1 |
| 36 | 4-4 | 128 | 128 | 4-4 | 128 | 128 | 8-8 | -1 | -1 |
| 37 | 4-4 | 128 | 128 | 4-4 | 128 | 128 | 8-8 | -1 | -1 |
| 38 | 4-4 | 128 | 128 | 4-4 | 128 | 128 | 8-8 | -1 | -1 |
| 39 | 4-4 | 128 | 128 | 4-4 | 128 | 128 | 4-4 | 128 | 128 |
| 40 | 4-4 | 128 | 128 | 4-4 | 128 | 128 | 4-4 | 128 | 128 |
| 41 | 4-4 | 128 | 128 | 4-4 | 128 | 128 | 8-8 | -1 | -1 |
| 42 | 4-4 | 128 | 128 | 4-4 | 128 | 128 | 8-8 | -1 | -1 |
| 43 | 4-4 | 128 | 128 | 4-4 | 128 | 128 | 8-8 | -1 | -1 |
| 44 | 4-4 | -1 | -1 | 4-4 | -1 | -1 | 4-4 | 128 | 128 |
| 45 | 4-4 | 128 | 128 | 4-4 | 128 | 128 | 8-8 | -1 | -1 |
| 46 | 4-4 | 128 | 128 | 4-4 | 128 | 128 | 8-8 | -1 | -1 |
| 47 | 4-4 | 128 | 128 | 4-4 | 128 | 128 | 4-4 | 128 | 128 |
| 48 | 4-4 | 128 | 128 | 4-4 | 128 | 128 | 8-8 | -1 | -1 |

Table 7: MxMoE W5A5 scheme (continued)

| Expert | Gate | | | Up | | | Down | | |
|--------|------|--------|--------|------|--------|--------|------|--------|--------|
| | w-act | w_gsize | a_gsize | w-act | w_gsize | a_gsize | w-act | w_gsize | a_gsize |
| 49 | 4-4 | 128 | 128 | 4-4 | 128 | 128 | 8-8 | -1 | -1 |
| 50 | 4-4 | 128 | 128 | 4-4 | 128 | 128 | 4-4 | 128 | 128 |
| 51 | 4-4 | 128 | 128 | 4-4 | 128 | 128 | 4-4 | 128 | 128 |
| 52 | 4-4 | 128 | 128 | 4-4 | 128 | 128 | 8-8 | -1 | -1 |
| 53 | 4-4 | 128 | 128 | 4-4 | 128 | 128 | 4-4 | 128 | 128 |
| 54 | 4-4 | -1 | -1 | 4-4 | -1 | -1 | 4-4 | 128 | 128 |
| 55 | 4-4 | -1 | -1 | 4-4 | -1 | -1 | 4-4 | 128 | 128 |
| 56 | 4-4 | -1 | -1 | 4-4 | -1 | -1 | 4-4 | 128 | 128 |
| 57 | 4-4 | 128 | 128 | 4-4 | 128 | 128 | 4-4 | 128 | 128 |
| 58 | 4-4 | -1 | -1 | 4-4 | -1 | -1 | 4-4 | 128 | 128 |
| 59 | 4-4 | 128 | 128 | 4-4 | 128 | 128 | 8-8 | -1 | -1 |
| 60 | 4-4 | -1 | -1 | 4-4 | -1 | -1 | 8-8 | -1 | -1 |

