# OpenReview forum: "MxMoE: Mixed-precision Quantization for MoE with Accuracy and Performance Co-Design"
_ICML.cc/2025/Conference — ICML 2025 poster_

### Official Review · Reviewer_sMX8 · 2025-02-20

**Overall Recommendation:** 3

**Summary:**

Mixture-of-expert LLMs are getting important to reduce the traning cost without sacrificing the final model quality. With more MoE LLMs released, it's important to optimize the serving performance for them. Quantization is an important technique to optimize the LLMs serving by use low-precision data types to store the parameters. The authors propose to use different quantization schemes (i.e., the low-precision data type for parameter and activation, and the group-size for channel-wise quantization) at the **linear-block** level instead of expert level. Besides, the authors also implement more efficient kernels to execute the quantized layers. Experiments show that the proposed method MxMoE achieves 2.4 lower Wikitext-2 perplexity than GPTQ at 2.25-bit and delivering up to 3.4x speedup over full precision.

**Claims And Evidence:**

Yes

**Essential References Not Discussed:**

No

**Experimental Designs Or Analyses:**

Yes

**Methods And Evaluation Criteria:**

Yes

**Other Comments Or Suggestions:**

No

**Other Strengths And Weaknesses:**

**Strengths**
- Provides an efficient implementation for quantized MoE layers.

**Weaknesses**
- The novelty of quantizing at the block level is incremental.
- Although the authors implemented new kernels to support the proposed quantized MoE layer, these contributions seem more like engineering improvements rather than groundbreaking innovations.

**Details**
The novelty of quantizing at the block level is incremental. As the authors mention in the background section, prior works like MC-MOE have already observed that different experts have varying levels of significance or importance and could be quantized into different low-precision data types. These works also employ linear programming to determine the optimal quantization strategy.

The new kernel implementation appears to be more of an engineering improvement. Since the quantization granularity is at the linear level, it is natural to parallelize different experts with different quantization schemes within a single kernel by using separate thread blocks to execute these layers or by utilizing different CUDA streams to launch distinct kernels. The innovations in this aspect seem more like engineering contributions to the community rather than fundamental algorithmic advancements.

The accuracy does not show significant improvement in the W3.5-A16 setting compared to the uniform baseline GPTQ. From Table 1, MxMoE (W3.5-A16) achieves similar average accuracy to GPTQ (W3.5-A16), and MxMoE only demonstrates better accuracy at even lower precision (e.g., 2.25-bit). If MxMoE or similar methods that use different quantized types for different experts only prove effective at very low precision (<3-bit), their impact and contribution would be considerably diminished.

**Questions For Authors:**

No

**Relation To Broader Scientific Literature:**

Efficient execution of quantized MoE LLMs.

**Theoretical Claims:**

No

---

> ### Author Rebuttal · Authors · 2025-03-31
>
> We appreciate your thoughtful feedback. We would like to provide further clarification on several aspects of our work that you have mentioned:
> 1. Novelty of the Approach and Similarities with Prior Work
>  - **Linear-Block-Level Quantization**: Our approach is driven by empirical observations rather than a straightforward extension of prior work. Figure 1(a) highlights significant sensitivity differences between linear blocks, and Table 3 confirms the effectiveness of our approach.
>  - **Objective: Balancing Model Accuracy and Hardware Efficiency**: Unlike prior work that solely optimizes model accuracy, MxMoE jointly considers both accuracy and system efficiency. Hardware-friendly quantization strategies improve efficiency but often degrade accuracy (Figure 6). Our objective function is designed to incorporate both quantization perturbation and the runtime cost of a mixed-precision scheme. And accurately modeling runtime cost is nontrivial, requiring deep analysis of hardware characteristics and parallel algorithm design, as detailed in Section 4.2.
>  - **Scope: Weight-Only vs. Weight + Activation Quantization**: MC-MoE focuses on extremely low-bit weight-only quantization, whereas MxMoE optimally allocates bit-widths for both activation and weight parameters. This approach enables MxMoE to leverage low-precision arithmetic units on GPUs more effectively (Figure 5).
>  - **Performance: Theoretical vs. Real Speedup**: Previous methods relied on existing kernels like HQQ, achieving only theoretical speedups without real-world reductions in wall-clock time (Figure 2). In contrast, MxMoE is designed from the ground up for hardware efficiency, leading to actual runtime improvements.
>  - **Automation: End-to-End Mixed-Precision pipeline**: MxMoE is the first framework to fully automate the generation of mixed-precision schemes and fused operators tailored to the generated scheme which is not achieved in any prior works.
> 2. MxMoE Is Engineering Optimization
> - MxMoE is not simply providing one kernel but rather offering an automated method for the mixed-precision allocation and generating customized fused operators. The design and optimization of mixed-precision GEMM have been an ongoing area of research[3][4], which continue to push the boundaries of quantization research. MxMoE addresses the more challenging problem of mixed-precision GroupGEMM, which is a much more complex problem that goes beyond simple parallelization of different precision tasks in thread-blocks or streams.
> - In fact, we initially implemented MxMoE with multi-stream, but unfortunately we found it could not fully exploit hardware. We show the comparison(RTX 4090, Qwen1.5, W4A16 and W8A8 mixed MoE-Block):
>
>   | #Tokens 1024|TFLOPS|
>   |-|-|
>   |FP16|70.117|
>   |W4A16|119.32|
>   |W8A8|140.95|
>   |MxMoE|163.20|
>   |multi-stream|142.19|
>
>   The results demonstrated that our approach achieves superior efficiency.
> - Assigning task to CTAs is fundamentally the minimum makespan problem, an NP-hard challenge. "parallelize different experts with different quantization schemes within a single kernel by using separate thread blocks" won't work as it suffers from workload imbalance. To address this, we design a tile scheduler that distributes workload evenly among all CTAs(Line 295).
> 3. Experimental Results
> - At 3.5 bit, quantization perturbation is relatively limited. Both GPTQ and MxMoE perform well at this setting. As the bit-width decreases, the accuracy advantage of MxMoE becomes more evident. This extreme low-bit compression is highly relevant for resource-constrained model deployment, as explored in several works such as [1][2].
> - MxMoE performs bit-width allocation for both activation and weight. As demonstrated in Table 1 and Figure 5, MxMoE shows a significant accuracy advantage over existing methods in weight-activation quantization, pushing the accuracy-efficiency Pareto frontier to new heights.
>
> In conclusion, we believe **MxMoE introduces novel contributions in both algorithmic design and system-level optimization**. We sincerely appreciate the reviewer’s feedback and the opportunity to clarify our contributions. We hope our responses have effectively addressed the concerns raised. Given these clarifications, we hope the reviewer may reconsider the overall evaluation of our work. We would be happy to further discuss any remaining questions or concerns.\
> [1] Ma, Shuming, et al. "The Era of 1-bit LLMs: All Large Language Models are in 1.58 Bits." CoRR (2024).\
> [2] Yuan, Zhihang, et al. "PB-LLM: Partially Binarized Large Language Models." The Twelfth International Conference on Learning Representations.\
> [3] Wang, Lei, et al. "Ladder: Enabling Efficient {Low-Precision} Deep Learning Computing through Hardware-aware Tensor Transformation." 18th USENIX Symposium on Operating Systems Design and Implementation (OSDI 24).\
> [4] Lin, Yujun, et al. "Qserve: W4a8kv4 quantization and system co-design for efficient llm serving." arXiv preprint arXiv:2405.04532 (2024).

---

> > ### Comment · Reviewer_sMX8 · 2025-04-04
> >
> > Thanks the authors addressing my concerns.
> >
> > I decide to change my score from 2 to 3.

---

> > > ### Author Response · Authors · 2025-04-08
> > >
> > > Thank you for your response and for increasing your score. We appreciate the time and effort you have spent providing valuable feedback.
> > >
> > > Best regards, Authors.

---

### Official Review · Reviewer_eUec · 2025-03-09

**Overall Recommendation:** 2

**Summary:**

This paper presents a framework for exploring the design space of mixed-precision quantization in mixture-of-experts (MoE) models. It considers the variation in quantization sensitivity of linear blocks within models, allocating different bitwidths based on sensitivity. Additionally, it takes into account the frequency of block activation to apply different quantization strategies. The proposed mixed-precision approach is claimed to achieve a speedup over uniform quantization while maintaining equivalent accuracy. The method is evaluated using DeepSeek, Mixtral, and Qwen MoE architectures, with comparisons across various quantization methods.

**Claims And Evidence:**

The primary claim is that the proposed mixed-precision quantization framework improves inference speed compared to uniform quantization at the same accuracy level. The claim is supported by experimental results demonstrating speedups on multiple MoE models. However, some key methodological aspects, such as how sensitivity and activation frequency incorporated into the quantization strategy, could be elaborated further. Additionally, the paper should provide more explanation of the quantization methods used in comparison, such as GPTQ and Quaro.

**Essential References Not Discussed:**

No.

**Experimental Designs Or Analyses:**

The experimental setup appears reasonable, but certain details are missing. Specifically, the paper describes the allocator, kernel generator, and task scheduler in Section 4.1 and Figure 1, but their implementation details remain unclear. Some information is presented in Section 4.3, but it does not sufficiently explain how these components function together. A more detailed breakdown of the experimental setup would be beneficial for reproducibility and understanding.

**Methods And Evaluation Criteria:**

The evaluation is based on well-known MoE models (DeepSeek, Mixtral, and Qwen) and considers several quantization techniques. The datasets used for evaluation appear appropriate, and the comparisons are reasonably extensive. However, more clarity is needed in describing how different quantization configurations are selected. Additionally, while the paper repeatedly mentions hardware-friendly quantization, it does not explicitly define what hardware efficiency entails—whether it refers to latency, throughput, or other tradeoffs.

**Other Comments Or Suggestions:**

* Clarify the definition of hardware-friendly quantization—does it prioritize latency, throughput, or another metric?
* Reduce redundancy in discussions of parameter sensitivity, activation frequency, and hardware characteristics. Instead, elaborate on how they are measured and affect performance.
* Provide clearer explanations of the quantization methods used in comparison.
* Expand the descriptions of DeepSeek, Mixtral, and Qwen MoE architectures, highlighting their relevance to this study.

**Other Strengths And Weaknesses:**

Strengths:
* The paper presents a useful and practical approach to improving MoE model efficiency through mixed-precision quantization.
* A good comparison is made between different MoE models and quantization schemes.
* The experimental results demonstrate performance improvements and validate the approach.

Weaknesses:
* The paper lacks clarity in several areas, including definitions of key terms (e.g., hardware-friendly quantization) and details on the allocator, kernel generator, and task scheduler.
* The related background on quantization methods (GPTQ and Quaro) and MoE models is insufficient.
* Some sections are repetitive, restating parameter sensitivity and activation frequency without elaborating on their exact implications.
* The experimental design could be explained in greater detail for better reproducibility.

**Questions For Authors:**

* Can you clarify what hardware-friendly quantization means in your context?
* Could you provide more details on the allocator, kernel generator, and task scheduler implementation?

**Relation To Broader Scientific Literature:**

The paper references several relevant works but lacks sufficient background discussion on some key concepts. Specifically, more context on GPTQ and Quaro, two quantization methods used in the experiments, would be helpful. Similarly, the features, differences, and challenges of the selected MoE models (DeepSeek, Mixtral, and Qwen) should be elaborated upon to provide a clearer motivation for their inclusion.

**Theoretical Claims:**

The paper does not present any theoretical claims, and therefore, there are no proofs to verify. Its contributions are primarily empirical.

---

> ### Author Rebuttal · Authors · 2025-04-01
>
> We appreciate the reviewer’s engagement.

---

### Official Review · Reviewer_CPad · 2025-03-12

**Overall Recommendation:** 3

**Summary:**

This paper presents MxMoE, a mixed-precision quantization framework for MoE models. The development of MxMoE is driven by three key factors:
1. Parameter Sensitivity,  linear blocks exhibit significant variability in their sensitivity to quantization.
2. Expert Activation Frequencies, activation frequencies of experts vary considerably across MoE blocks.
3. Hardware Characteristics, the weight-activation quantization scheme can be optimized to align with its actual arithmetic intensity for enhanced performance.

## update after rebuttal

I asked some questions regarding the rebuttal, and the authors' response makes sense. I have maintained my score.

**Claims And Evidence:**

yes

**Essential References Not Discussed:**

N/A

**Experimental Designs Or Analyses:**

To the best of my knowledge in this field, most of the experimental designs or analyses are sound. I still have concerns regarding why MxMoE uses the 5-5 setting in Table 1 for weight-activation quantization. Could the observed improvements simply be attributed to the 1-bit increase in precision? I would appreciate it if the author could clarify the rationale behind that.

**Methods And Evaluation Criteria:**

The benchmarks and evaluation criteria are widely used to assess the performance and computational efficiency of quantized models. For instance in baseline method, QuaRot [1].
To thoroughly evaluate computational throughput, we recommend that the authors incorporate additonal benchmarks with varied length distributions.

[1] https://proceedings.neurips.cc/paper_files/paper/2024/hash/b5b939436789f76f08b9d0da5e81af7c-Abstract-Conference.html

**Other Comments Or Suggestions:**

There are some typos for the user's further improvements, such as the incorrect use of capitalization in line 194 and Figure 3.
Captions can be further refined; for instance, directly presenting a configuration like g128 and g-1 is not reviewer-friendly for those unfamiliar with the field.

**Other Strengths And Weaknesses:**

N/A

**Questions For Authors:**

[1] Why MxMoE uses the 5-5 setting in Table 1 for weight-activation quantization. Could the observed improvements simply be attributed to the 1-bit increase in precision? (See Section "Experimental Designs Or Analyses")

[2] Can the authors incorporate additional benchmarks with varied length distributions to conduct throughput-related tests? （See Section Methods And Evaluation Criteria）

**Relation To Broader Scientific Literature:**

I have not come across any papers that share the key contributions of this one.

**Theoretical Claims:**

This paper does not contain proofs for theoretical claims.

---

> ### Author Rebuttal · Authors · 2025-04-01
>
> We sincerely appreciate your recognition of our work. Your insightful questions and suggestions have been instrumental in enhancing the quality of our manuscript. Below, we address your concerns in detail:
> 1. Rationale for the W5A5 Configuration in Table 1
>   - **Why does the "strange" W5A5 setting arise?**
>
>     This configuration is intentional. During bitwidth allocation, we set an average 5-bit activation and weight budget for MxMoE. On our experimental platform (RTX 4090), Tensor Cores support 4-bit and 8-bit operations, allowing both W4A4 and W8A8 configurations to benefit from low-precision Tensor Core acceleration. The additional 1-bit budget provides MxMoE with greater flexibility in bitwidth allocation, meaning that certain activations are assigned 8-bit precision while others remain at 4-bit.
>
>     Although the average bitwidth increases by only 1 bit, MxMoE automatically identifies activations that are more sensitive to quantization through sensitivity analysis and assigns them a higher bitwidth to mitigate accuracy degradation. This exemplifies the advantage of mixed-precision quantization over uniform quantization.
>
>   - **Does the accuracy gain stem solely from the additional 1-bit?**
>
>     Yes. The improved accuracy of W5A5 over W4A4 is primarily due to extreme outliers in the input activations of the down_proj layer—a phenomenon well-documented in prior research (e.g., [1, 2]). Unlike standard outlier features, these activations are particularly difficult to quantize effectively at 4 bits, leading to precision loss. MxMoE mitigates this issue by allocating higher bitwidths to such activations, thereby preserving model accuracy.
>
>     Our findings align with prior work [2], which demonstrates that suppressing extreme outliers substantially improves post-quantization model performance. We will clarify this rationale in the revised manuscript and provide a case study on bitwidth allocation.
>
> 2. Throughput Evaluation with Varied Length Distributions
>
>     We agree that evaluating computational throughput across diverse input lengths is essential for real-world applicability. In response to your suggestion, we have expanded our throughput experiments. Due to time constraints, we conducted experiments on DeepSeekV2-Lite using the Humaneval-X dataset.
>
>     In this study, we employed the exact same MxMoE(W5A5) configuration as in Table 1 and Figure 5. To assess performance across varied sequence lengths, we set the batch size to 1—ensuring that input lengths is varied, unlike in the compute-bound scenarios in Figure 5. We then processed the entire dataset, computing the average computational throughput (total FLOPs divided by execution time) and comparing it to FP16 performance. The length distribution of the dataset can be found at https://huggingface.co/datasets/THUDM/humaneval-x.
>
>     | Dataset     | FP16  TOPS | MxMoE(W5A5) TOPS | Speedup |
>     | ----------- | ----- | ----------- | ------- |
>     | humaneval-x | 38.19 | 107.36      | 2.8     |
>     ||||
>
>     As shown in the results, although the speedup is slightly lower than in the fully compute-bound scenario, MxMoE(W5A5) maintains a strong acceleration ratio even on datasets with varied-length short sequences. This highlights the robustness of the mixed-precision approach.
>
>     We will expand this experiment in the revised manuscript and add more models and dataset tests. Thank you very much for your suggestions!
>
> 3. Typos and Figure Refinements
>
>     We sincerely appreciate your careful review and apologize for the typographical errors. These will be corrected in the revised manuscript. Additionally, we will refine the caption of Figure 3 to explicitly define terms such as g128 (quantization group size of 128) and g-1 (per-channel/token quantization) to enhance clarity. Thanks again for your advice!
>
> Once again, we deeply appreciate your valuable feedback. Your suggestions have been instrumental in improving the quality of our manuscript. If you have any further suggestions or concerns, we would be delighted to discuss them!
>
>  [1] Sun, Mingjie, et al. Massive Activations in Large Language Models. First Conference on Language Modeling.\
>  [2] Lin, Haokun, et al. Duquant: Distributing Outliers via Dual Transformation Makes Stronger Quantized LLMs. NeurIPS 2024.

---

> > ### Comment · Reviewer_CPad · 2025-04-02
> >
> > Thank you for your rebuttal. I have reviewed both the rebuttal and the paper for some time. It seems there may be some misunderstanding regarding my question. I was asking whether the performance gains over the baselines are due to the additional 1-bit precision. If your answer is yes, and you acknowledge that a higher bit width for activation is crucial, could you please include another heterogeneous quantization strategy while maintaining the 5-5 setting to help convince me of MxMoE’s effectiveness? This would allow me to maintain a positive overall assessment. Thanks again.
> >
> >
> >
> > ### Update:
> >
> > The reply makes sense to me. Thanks to all the authors for their efforts. I kept my score positive.

---

> > > ### Author Response · Authors · 2025-04-02
> > >
> > > Thank you for your valuable feedback. In response, we conducted two experiments.
> > >
> > > Firstly, we test Qwen1.5-MoE with the model perplexity on wikitext-2 (sequence length: 4096) to analyze the effects of various quantization bit settings (RTN per-channel/token symmetric quantization).
> > >
> > > For reference, full precision perplexity is **6.791**.
> > >
> > > The results are shown in the table below (rows: weight bits; columns: activation bits):
> > > Perplexity (Lower is Better)
> > >
> > > | #Bits (W-ACT) |     4     |    5   |    6   |   7   |   8   |
> > > |:-------------:|:---------:|:------:|:------:|:-----:|:-----:|
> > > | 4             | 68079.039 | 41.433 | 11.298 | 9.406 | 8.068 |
> > > | 5             | 12305.585 | 38.707 | 9.715  | 8.169 | 7.335 |
> > > | 6             | 14251.822 | 26.297 | 9.216  | 8.196 | 7.204 |
> > > | 7             | 18151.474 | 34.775 | 9.747  | 8.182 | 7.325 |
> > > | 8             | 19091.917 | 38.99  | 9.525  | 8.26  | 7.278 |
> > >
> > > Observation: Increasing the activation bitwidth dramatically reduces the perplexity while the benefits of increasing the bitwidth of weight are comparablely margin. In particular, **moving from 4-bit to 5-bit activations results in significant performance improvement**. While further increasing the bitwidth beyond 5 bits continues to lower the perplexity, the gains become less pronounced.
> > >
> > > The experimental results in the table confirm the effectiveness of heterogeneous quantization strategy.
> > >
> > > To better illustrate the effectiveness of MxMoE, we conducted an additional experiments (Qwen1.5-MoE). In below experiments, both Quarot and MxMoE were evaluated under RTN-quantized weight settings (fully aligned with the Quarot-RTN configuration in [1]. This decision was made because GPTQ is too slow for processing weights).
> > > The performance of Quarot under various precision(uniform quantization) settings is summarized below:
> > >
> > > |   Setting  |  w4a4  |  w5a5 | w6a6 |  w7a7 |  w8a8 |
> > > |:----------:|:------:|:-----:|:----:|:-----:|:-----:|
> > > | Perplexity | 36.385 | 7.998 | 6.99 | 6.852 | 6.814 |
> > >
> > > MxMoE(W5A5-RTN): **7.160**
> > >
> > > Conclusion:
> > > 1. **Quarot also shows a marked improvement when moving to a W5A5 configuration compared to W4A4**
> > > 2. Quarot still underperforms relative to MxMoE. This difference is attributable to MxMoE's ability to **identify sensitive parameters and allocate higher precision to protect them (e.g. 8-bit for sensitive activations, 4-bit for non-sensitive activations)**, thereby better mitigating the quantization perturbation.
> > > 3. MxMoE achieves wall-clock time reduction under the W5A5 setting through mixed-precision strategy while **uniform W5A5, W6A6, W7A7 quantization schemes are not supported by modern hardware(e.g. GPU)**.
> > >
> > > We sincerely appreciate your prompt, thoughtful feedback and particularly value your insightful observation about methodological details. In the revised version, we will include an in-depth case study that discusses these experimental results and further illustrates the effectiveness of MxMoE.
> > >
> > > Thank you again for your constructive feedback and for engaging in this detailed review process. We are happy to provide any additional information or experiments to address any remaining concerns!
> > >
> > > [1] https://proceedings.neurips.cc/paper_files/paper/2024/hash/b5b939436789f76f08b9d0da5e81af7c-Abstract-Conference.html

---

### Official Review · Reviewer_ELGx · 2025-03-13

**Overall Recommendation:** 3

**Summary:**

The paper introduces MxMoE, a mixed-precision quantization framework tailored for Mixture-of-Experts (MoE) models, aiming to address deployment challenges posed by their large memory footprint and computational demands. Key insights include:
1. Heterogeneous quantization sensitivity: Linear blocks within MoE experts exhibit varying sensitivity to bitwidth reduction.
2. Divergent expert activation frequencies: Computational characteristics (e.g., memory-bound vs. compute-bound operations) differ across experts.
MxMoE optimizes bitwidth allocation at the linear-block granularity (rather than expert-level) and generates specialized GPU kernels for parallel execution of mixed-precision Group-GEMM operations. It formulates the problem as an Integer Linear Programming (ILP) task, balancing quantization-induced accuracy loss and runtime efficiency. Evaluations on models like DeepSeek-V2-Lite and Mixtral-8×7B show MxMoE achieves 2.4× lower perplexity than GPTQ at 2.25-bit and 3.4× speedup over full-precision execution, outperforming uniform quantization baselines.

**Claims And Evidence:**

Yes.

**Essential References Not Discussed:**

Not sure.

**Experimental Designs Or Analyses:**

There is a lack of validity analysis experiment for each part of the method, especially Micro-Kernel Specialization and Resource Configuration.

**Methods And Evaluation Criteria:**

The trade-off parameter $r$ requires tuning for different models and use cases, which is not thoroughly explored.

**Other Comments Or Suggestions:**

No.

**Other Strengths And Weaknesses:**

1. Novel granularity: Linear-block-level quantization allocation improves accuracy-efficiency trade-offs compared to expert-level or uniform approaches.
2. Hardware-algorithm co-design: Integrates parameter sensitivity, activation patterns, and hardware constraints (e.g., roofline model) into a unified framework.
3. Performance gains: Demonstrates significant improvements in perplexity and throughput across multiple models and workloads (memory/compute-bound).

**Questions For Authors:**

Please see "Experimental Designs Or Analyses" and "Methods And Evaluation Criteria"

**Relation To Broader Scientific Literature:**

Not sure.

**Theoretical Claims:**

No theoretical claims.

---

> ### Author Rebuttal · Authors · 2025-04-01
>
> We sincerely appreciate the reviewer’s constructive feedback. Below, we address the two key concerns raised:
>
> 1. Exploration of Trade-off Parameter $r$ for Different Models and Use Cases
>
>     Our ablation study (Figure 6) explores the impact of the hyper-parameter $r$ on model accuracy and hardware efficiency. Our analysis demonstrates that increasing $r$ prioritizes accuracy (e.g., lower perplexity) at the expense of efficiency (e.g., lower throughput).
>
>     **MxMoE can automatically adjust $r$ according to use case**: there are typically some constraints such as memory budget, precision budget, and latency budget in model quantization. MxMoE satisfy these constraints by automatically adjusting $r$. Additionally, users may manually tune $r$ to balance hardware efficiency and model accuracy, selecting a suitable value based on their specific requirements. Basically, if a user prioritizes higher accuracy, $r$ should be set closer to 1, whereas if better speed is desired, $r$ should be closer to 0.
>
>     While Figure 6 presents results for a single model, we emphasize that the observed trend is consistent across all evaluated architectures. **For simplicity and reproducibility, we adopt $r=0.75$ as the default value in all experiments, except for low-bit weight-only quantization**, where we set $r=1.0$ (Lines 363, 373, 378). This exception aligns with edge deployment scenarios, where memory constraints dominate, making it crucial to maximize compression while preserving accuracy.
>
> 2. Validity Analysis for Method Components
>
>     **Micro-Kernel Specialization**
>
>     MxMoE mitigates the combinatorial explosion of mixed-precision configurations by automating kernel generation (Line 289). To illustrate the effectiveness of our approach, we compare our strategy with two alternative solutions:
>
>     - **Developing a universal kernel to handle all precision combinations**: This approach would compromise kernel performance. We provide a breakdown demonstrating the limitations of this method relative to our micro-kernel specialization. Specifically, the kernel for W4A4-per-channel could theoretically share the same software pipeline with W4A4-group128, but enforcing universality significantly degrades performance (test shape $[8192, 8192, 8192]$):
>
>         | Kernel Type | W4A4_per-channel TOPS | W4A4_group128 TOPS |
>         |-| - |-|
>         | W4A4_per-channel(Specialized) | 1070.5303| N/A  |
>         | W4A4_group128(Specialized)| N/A  | 667.3349|
>         | Unified Kernel| 929.1997 | 412.0268  |
>         ||||
>
>         Unifying the two pipelines requires introducing runtime condition checks, which hinder loop unrolling in the MAC-loop. Moreover, to support group-size=128, the per-channel kernel’s tile-size selection is constrained, making configurations such as tile_k=256 infeasible.
>
>     - **Developing separate kernels for each configuration**:
>
>         While handcrafted kernels could match performance, they require substantial engineering effort. If a given hardware platform supports five quantization candidates (e.g., w2a6, w4a16, w8a8, w4a4, w4a4 with group-size 128), implementing individual kernels for all possible configurations would require $5!=120$ kernels! In contrast, our micro-kernel specialization approach requires implementing only 5 configurable micro-kernels, which are automatically combined by the kernel generator to form optimized fused operators.
>
>     **Resource Configuration**
>
>     To resolve resource mismatches caused by heterogeneous micro-kernels (e.g., varying warp counts per thread block, as shown in Figure 4), we employ the following strategies:
>
>     - Max-resource allocation: Ensuring all parallel units reach the resource ceiling for correctness.
>
>     - Slice-K optimization: This optimization improves shared memory utilization and achieves speedup compared to a baseline without carefully config, we provide a case study: The W4A16-per-channel quantization with/without resource configuration optimization(test shape $[16, 8192, 8192]$).
>
>         | Method | Best Config| W4A16 TOPS |
>         | -- | - | -- |
>         | $\mathrm{MxMoE}$   | Tile: [16, 64, 128]; Warp: [1,2,2]  | 74.8983    |
>         | $\mathrm{- Resource Configuration}$ | Tile: [16, 128, 128]; Warp: [1,4,1] | 52.4288    |
>         ||||
>
>         The result shows that with resource configuration optimization, MxMoE automatically discovered a better tile configuration (assign 2 warps along k-dimension, which is not a common configuration in previous kernel such as Marlin[1] e.t.c.), which brings 42% speedup.
>
> We will enhance the revised manuscript with additional ablation studies and quantitative comparisons to reinforce these claims. We sincerely appreciate the reviewer’s valuable suggestions, which we believe further strengthen the technical rigor and reproducibility of our work.
>
> [1] Frantar, Elias, et al. "Marlin: Mixed-precision auto-regressive parallel inference on large language models." Proceedings of the 30th ACM SIGPLAN 2025.

---

### Decision · Program_Chairs · 2025-05-01

**Decision:**

Accept (poster)

**Comment:**

This paper proposes a new MoE model quantization method with mixed precision considering the sensitivity of the linear layers considering various factors. Also, the proposed MxMoE automatically generates optimized Group-GEMM kernels for mixed precision.

In the review process, there were several clarification questions from the reviewers including trade-off parameter r, micro-kernel specialization, usage of W5A5 and novelty of the proposed method. In the rebuttal and discussion phase, the authors provided relevant answers and information to those questions.

After the discussion, all the reviewers excluding the one detected as LLM-generated had a consensus of weak accept. (3 weak accepts) In the reviewer discussion, a reviewer shared that the proposed method made sense, but was not exciting.

Combining all those information, I recommend a weak accept if there's an available spot for this paper as it potentially has some contributions even though it may not be significant.